# Network Organization of Antibody Interactions in Sequence and Structure Space: the RADARS Model

**DOI:** 10.3390/antib9020013

**Published:** 2020-05-06

**Authors:** József Prechl

**Affiliations:** Diagnosticum Zrt., 126. Attila u., 1047 Budapest, Hungary; jprechl@diagnosticum.hu

**Keywords:** antibody, network, sequence, structure, clonality, B cell, systems biology, quantitative biology, immunodominance, consecutive reactions

## Abstract

Adaptive immunity in vertebrates is a complex self-organizing network of molecular interactions. While deep sequencing of the immune-receptor repertoire may reveal clonal relationships, functional interpretation of such data is hampered by the inherent limitations of converting sequence to structure to function. In this paper, a novel model of antibody interaction space and network, termed radial adjustment of system resolution, RAdial ADjustment of System Resolution (RADARS), is proposed. The model is based on the radial growth of interaction affinity of antibodies towards an infinity of directions in structure space, each direction corresponding to particular shapes of antigen epitopes. Levels of interaction affinity appear as free energy shells of the system, where hierarchical B-cell development and differentiation takes place. Equilibrium in this immunological thermodynamic system can be described by a power law distribution of antibody-free energies with an ideal network degree exponent of phi square, representing a scale-free fractal network of antibody interactions. Plasma cells are network hubs, memory B cells are nodes with intermediate degrees, and B1 cells function as nodes with minimal degree. Overall, the RADARS model implies that a finite number of antibody structures can interact with an infinite number of antigens by immunologically controlled adjustment of interaction energy distribution. Understanding quantitative network properties of the system should help the organization of sequence-derived predicted structural data.

## 1. Introduction

Appearance of complex multicellular life was accompanied by the evolution of mechanisms that maintain cellular and molecular integrity in the host organism, called immunity. In vertebrates, mechanisms of immunity took to a new level by acquiring genetic, anatomical, and cellular features that together formed a new system [1]. The adaptive immune system is a complex system in the physical sense, being composed of a vast number of cells that engage in interactions, self-organized and, most impressively, adaptable to the molecular and cellular environment. Its mere size, with cell numbers in the range of 10^11^ in humans, suggests that the system could be described by statistical properties. In fact, the host is more than an organism, a supraorganism [2,3], with microbial communities, and immunity maintains a continuity of interactions in a wide range of energy scale, rather than simply discriminating self from nonself [4]. Technological advances now allow us to measure and characterize this complexity in ever-growing details, at the gene, transcript, protein, and cellular levels, driving the field of systems immunology [5]. The vast amount of data generated requires not only data storage and analysis capacity but also theoretical frameworks and models that simplify data organization and systems-level interpretation.

Humoral adaptive immunity comprises the cells and mechanisms that lead to the production of antibodies. In human adults, B cells develop in the bone marrow throughout life and build up a system of effector and memory cells, which accumulate with a lifetime of immunological experiences. The primary role of B cells and antibodies is the regulation of antigen removal and thereby adjustment of antigen concentrations in the host, operating over several orders of magnitude [6]. Because this specificity is genetically coded in the individually rearranged and mutated immunoglobulin heavy and light chain sequences, it is possible to capture the antibody repertoire in a given sample of B cells. Deep sequencing or next-generation sequencing (NGS) is capable of generating sequence data of antibody repertoires with varying resolution and length [7,8,9,10,11]. It is also possible to profile the antibody repertoire functionally, based on the identification of antibodies binding to huge sets of potential targets [12,13]. Antigen microarray assays are useful for the focused analysis of antibodies related to allergy, autoimmunity, infection, or cancer [14,15,16,17,18]. This approach is biased by the fact that a priori knowledge of targets is not always possible and only those antibodies that bind to the tested antigens are identified. High-throughput screens of proteomes for antigens [19] or of random peptide libraries for peptidic epitopes [20] can identify immunogenic targets. Such functional analyses provide a more meaningful profile in the immunological sense and, if carried out from blood, it is less prone to sampling error than cell-based sequencing approaches.

The relationship between antibody sequence and structure is on one hand like that of proteins in general, polypeptide chains of a given sequence fold into structures, which are responsible for function. In the enormous sequence space allowed by permutating amino acids, only the thermodynamically stable structures materialize as proteins [21]. Proteins capable of interacting with molecules in a way that improves chances of survival of the host organism will themselves survive and evolve. On the other hand, unlike proteins in general, antibodies evolve within the lifetime of the host. Importantly, because there are so many more sequences than structures and because changing a few critical amino acids can result in different structures, mapping sequence space to structure space is far from trivial. The combined length of the complementarity determining regions (CDR) of heavy and light immunoglobulin chains, which forms the antigen binding site, is around 14–50 amino acids (IMGT definition [22]), which allows extensive diversification. By employing screening and selection mechanisms, coupled with cycles of random mutagenesis, targeting primarily these amino acids, the immune system is capable of developing high-affinity antibodies against most targets. Understanding these processes on the systems level preferably requires the prediction of structures from NGS data [23] because of the complex sequence-to-structure relationship, as noted above.

The architecture and functioning of complex systems can be assessed by network science, which in the case of antibodies identifies antibody–antigen interaction networks [24]. The development of concepts of the immune system as a network were key steps in our current perception of immunity [25,26]. Efforts are now under way to describe the immune system as a network (termed network systems immunology) using NGS data and network science [27]. Since the system is organized by structure rather than sequence, the conceptualization of an antibody interaction network based on physical properties, like energy and probability of interaction, should help better define the system by integrating it into sequence-based structure prediction datasets.

In this paper, following a brief introduction to the sequence space of antibodies, a model for the molecular organization of antibody structure space and interaction space is proposed. The model builds on the concept and assumptions of the generalized quantitative model of antibody homeostasis [6,28,29], thus approaches antibody function from the physicochemical perspective, i.e., antibodies are organized into a network based on their binding affinity to cognate target. The model also considers the architecture of B-cell development and hierarchy and provides a power law-based quantitative network description of the humoral immune system in global equilibrium.

## 2. Antibody Clonal Representation in Sequence Space

Sequence space in the biological sense is a theoretical space comprising collections of nucleic acid or protein sequences of interest. We usually talk about protein sequence space and define what protein sets are involved (proteome of a given species, cells, etc.) and whether any restrictions are applied to the set (fully random, functional, identified, etc.). An amino acid sequence with a given length “d” and full randomization with 20 amino acids occupies a sequence space with size 20^d^ (Figure 1a). Since exponential growth is incredibly fast, longer sequences lead to the generation of vast amounts of sequence space in high dimensions. An exact sequence with no ambiguity defines an exact position in sequence space; moves in this space are discrete steps in a given dimension. It is accepted that only a fraction of all theoretically possible sequences are thermodynamically stable and protein evolution can be interpreted as a search for acceptable and functional structures in sequence and structure space [21]. Thinking along these lines, the evolution of antibody-binding surface, the paratope, is a search for the thermodynamically stable sequences and the selection from among these the ones meeting immunological criteria for B-cell survival. The set of viable antibody sequences, functional antibody sequence space, lies much below the theoretically possible [30] and close to the already observed and annotated antibody sequence space [31].

Collections of antibody protein sequences obtained by translating DNA or RNA of deep sequencing data ideally span the whole variable domain of heavy (VH) and light chains (VL) and can also pair these two. In such a case, the gene segments contributing to the rearrangement of VH and VL can be predicted and visualized in 3D and 2D, respectively, as shown in Figure 1. A repertoire can be represented by identifying coordinates of rearrangements identified, and symbol size or color can represent segment frequencies [32]. While the use of gene segments for classification allows tremendous reduction in dimensionality, it is not best suited for functional network analysis, where the use of complete rearranged and mutated sequences is preferable [33].

In a simpler approach, heavy-chain CDR3 regions only are used as an estimate of diversity. This region is regarded as being most important for determining binding specificity, but some precaution is necessary when only H-CDR3 is used for functional classification [34]. Selection of the pre-BCR bearing cells depends on signals that may be triggered by ubiquitous ligands present in the bone marrow microenvironment. The presence of uniform reactivity against such common public self-antigens may lead to the positive selection of CDR3 with similar binding properties and thereby similar sequences. Sequencing of the complete heavy-chain variable domains can be readily used to follow changes in repertoire size and diversity during B-cell development and response to immunization [35]. Whatever the depth and methodology, sequence similarity relationships can be used for the construction of family trees, often displayed in circular forms. These trees usually start classification with the V segment, clustering clones with common V use [36]. While this approach may be useful for classification, the use of the complete VDJ-H sequence as a first stage classifier, followed by VJ-L use, better reflects the natural development scheme of B cells (Figure 1c). Antibody repertoire sequencing now follows guidelines to help the integration of data [37,38,39]; several tools and databases devoted especially for these data have been established [8,10,40,41,42].

## 3. Antibody Interaction Representation in Structure Space

In contrast to this graded qualitative scheme, which may well serve the purpose of tracking peripheral clonal expansions accompanied by affinity maturation, a quantitative scheme should place genetic changes into structure rather than sequence space. Furthermore, because it is not only just the antibody structure but also the availability of targets and the structure of those targets that determine the development of antibody repertoire and the architecture of the network, we shall talk about interaction space, as explained below.

### 3.1. Structural Resolution of Molecular Recognition as a Measure of Interaction Strength

While sequences can be defined with various levels of certainty of an amino acid occupying a given position in the sequence, a static molecular structure can be defined at various levels of resolution. As we are talking about antibody molecules, structural resolution is on the atomic scale and crystal structures define atomic coordinates on the Ängstrom scale. The binding site of an antibody can also be characterized by the surface area that comes into close contact with the antigen [43,44]. Water molecules are displaced from this area as a function of the goodness of fit. The so-called buried surface area (BSA) is therefore a predictor of binding energy of protein interactions [45,46]. Another measure of goodness of fit is the decrease of free energy of the antibody molecule upon binding. All these approaches are correlated: higher resolution “recognition” of a structure by the antibody corresponds to greater BSA and to a higher binding energy. In other words, the resolution of molecular recognition is the goodness of fit in terms of number and strength of noncovalent bonds forming between antibody and target and can be expressed as free energy change ΔG or as equilibrium constant K of binding. These two are related by the equation
ΔG = ^−^k_B_TlnK(1)
Alternatively,
K = exp(^−^ΔG/k_B_T)(2)
where kB is the Boltzmann constant, which serves to relate the energy of particles to temperature, and T is thermodynamic temperature in Kelvins. Considering that the recognition of a given antigen is changing in the system by the adjustment of Ab fit, we can characterize this maturation of affinity by examining the change of free energy of the antibody upon binding, such as
ΔG = F_b_^−^ F_f_(3)
where F_b_ is the free energy of the bound (also called native), F_f_ is that of the free (nonnative) form of the antibody, and ΔG is the Gibbs free energy of binding.

The advantage of using thermodynamic description for the characterization of structural resolution is that it conveys the sense of function: higher binding energy means higher affinity of antibody to target, which, in turn, means more efficient clearance (6). Besides defining resolution of molecular recognition, which is a general descriptor, the identification of a given interaction requires the description of target shape, a distinct molecular structure. The higher the resolution, the more information is required for defining shape, corresponding to higher interaction energy and a better fit between antibody and target. This fit can be characterized by the number of bonds forming between pairs of atoms, which we shall denote by “r,” and by the fluctuation in energy of coupling ∆E^2^. The statistical distribution of affinity, based on random energy models [47,48] and protein folding models [49], that are also used for modeling BCR energy landscape [50], is a normal distribution of free energy ∆G around its mean <∆G> [51], according to
f(∆G) ~ exp(^−^(∆G^−^<∆G>)^2^/2r∆E^2^)(4)

By increasing resolution, we shall be able to distinguish between different shapes, the higher the resolution, the more shapes becoming distinct. Because of the structural complexity of an antibody-binding surface, the distinction between all possible shapes at high resolution would require a huge multidimensional space. Let us instead gradually generate an interaction space by considering a point of origin, the center of the system, from which a particular direction corresponds to a particular shape. In this representation, the extent by which we leave the point of origin corresponds to the resolution at which we can define the direction. Thus, going away from the minimal resolution, we can define shape at gradually higher resolutions, corresponding to larger free energy decrease of the interacting molecule (Figure 2a,b). Different levels of resolution, i.e., different levels of binding energies, appear in our scheme as shells of a sphere. Theoretically, the number of directions originating from a single point is infinite, so the shapes available in this representation could also be infinite. Practically, considering a reversible interaction, the resolution is limited by the binding energy of reversible interactions and system size, which is the number of available B cells.

This model of the organization of interactions of a system we shall call “RAdial ADjustment of System Resolution” or RADARS in short. The abbreviation intentionally reminds of radiolocation, where emitted electromagnetic waves interact with objects in their way and are reflected to provide an image of the surroundings. The RADARS model implies that elements of the growing system interact with the surroundings, gaining information and adjusting system growth accordingly.

### 3.2. B-Cell Development in Interaction Space

Immunological interpretation of the model requires us to fit B-cell development and antibody network growth into this interaction space. We shall assume that a common lymphoid progenitor (CLP) has the potential to generate any and all functional VDJ-VJ sequences and therefore to produce via sequential differentiation and maturations steps antibody against any and all targets. By functional VDJ-VJ sequences, we mean all sequences that are physically and biologically viable. This means thermodynamic stability (able to fold into a structure compatible with the Ig domain), ability to pair, forming a VH-VL functional binding unit, and ability to sustain a B cell via delivering survival, differentiation, and proliferation signals [28,52]. Of note, the value of r is 0 for a CLP and therefore the probability of interactions is 0, according to Equation (4), which confirms its position at the core of the system.

A differentiation step increases this total potential but immediately introduces restrictions in shape space because it has a direction. This will appear as a step towards increased resolution of cognate target structure recognition. Expression of the surrogate light chain (SLC) marks the first step towards B-cell antigen receptor (BCR) formation. These pro-B cells represent the founders of all B cells (Figure 3). While signaling via the SLC may be required, it is uncertain whether binding of SLC is required for further differentiation; therefore, we assume that these cells seed the complete antibody interaction space. Rearrangement of the heavy chain introduces a structural restriction: a particular functional heavy-chain variable domain (VH) sequence has a limited range of targets, a direction in shape space. Pre-B cells displaying the pre-BCR composed of VH-SLC pairs will divide until and as long as antigenic ligands are available. Cells with different VH sequences will populate the interaction space and share this space according to the availability and direction of target. Cells with more abundant targets expand more, cells with less frequent targets remain in lower numbers, until optimal BCR engagement is achieved [28]. As a result, interaction space as represented at these energies will be filled with different pre-B cell clones according to the availability of the common self-antigens. The next levels of interaction resolution, introducing further focusing in interaction space, comes with the rearrangement of the light chain. Individual pre-B cells rearrange their light chains independently and randomly. Therefore, all pre-B cells reserve a particular area on the next level of interaction energy. The size of this area again will correspond to the nature of the rearranged VL domain, with those finding more available targets having higher chance of survival. The pool of immature B cells thus fills the outer energy level in the bone marrow (Figure 3).

Taking a somewhat unique route of differentiation are the B1 cells. These cells seem to generate antibodies that keep B1 cells in a continuous state of low-level activation. This may reflect their ability to respond to soluble, highly abundant antigens [28], or an intrinsic ability of the BCR of sustained signaling, perhaps due to structural properties. In any case, B1 cells represent a stable population with the highest affinity towards self and nonself, which is achieved without affinity maturation. Meanwhile, B2 cells are continuously generated but die in a few days unless recruited as effector cells for further differentiation (Figure 3).

As implied above, selection in the bone marrow is a passive process: randomly generated sequences find their positions in interaction space, expanding or dying according to the availability of interacting target structures. As a result, the emerging population of immature B cells will bear the low-resolution antigenic signature of the bone marrow environment. This can be interpreted both as deletion of highly self-reactive clones to prevent autoimmunity [52] and as selection for mildly autoreactive clones to help homeostatic antibody functions and setting a reference for recognition of nonself [53]. From our point of view, cells with functional BCR and a given probability of engagement of BCR emerge from the bone marrow.

### 3.3. Immune Responses in Interaction Space

The development of cells reacting with antigens that are only temporarily present in the host presents the risk of investing energy into clones that will become useless once that antigen disappears. Therefore, such clonal expansions beyond the border of self only take place when second signals inform the host of danger. This is the development of an immune response, aided by various molecular and cellular help signals. Thymus-independent and primary thymus-dependent responses expand populations of B cells without improving their affinity, thus keeping them on the same level in the interaction space. Thymus-dependent responses aided by helper T cells lead to the formation of germinal centers (GC) where affinity maturation takes place. This adjustment of affinity by repeated cycles of random somatic mutation and selection focuses interactions into a particular direction in antigenic structure space, leading to the growth of system only in that direction. Postgerminal center B cells will have accumulated somatic hypermutations to increase their affinity. This corresponds to increased values of r, the number of noncovalent bonds in the binding surface. In this region of interaction space, most effector cells are eliminated by apoptosis and some cells go into resting state, once target is cleared. These latter are the memory B cells (MBC) that conserve the genetic information acquired during affinity maturation but minimize their activity, i.e., no divisions and no antibody secretion, remaining guards against future encounter with the same or similar target (Figure 2) [54,55,56]. Another important type of cell that remains in the system is the long-lived plasma cell (LLPC), which is a terminally differentiated B cell that has become independent of BCR signals and is fully devoted to antibody secretion [57,58].

## 4. Characterization of the Antibody Interaction Network

### 4.1. Distribution of Binding Energies in the System

Cell and systems biological aspects of the RADARS model are summarized in Figure 4. Expansion and differentiation of cells originating from CLPs create a cellular repertoire in the bone marrow; then, further expansions and differentiation steps increase this repertoire and supplement it with antibody secretion. The development of B cells bearing surface antibodies, B-cell receptors, with increasing affinities takes place in an environment loaded with a huge diversity of macromolecules. These antibodies thus develop in a system characterized by constant reversible, noncovalent interactions. These interactions in the system can be described mathematically by the frequency distribution of interaction energy. Based on the random energy model, the free energy of biomolecular interactions is normally distributed [51].

Testing a given antigen against a universe of randomly generated antibodies would yield a normal distribution of binding energies. However, it is exactly the role of the immune system to adjust the binding energy against particular antigens according to their quality of selfness and dangerousness. The Generalized Quantitative Model of antibody homeostasis proposes that antigen concentrations can be best regulated in the host by setting the equilibrium dissociation constants nearly equal to the desired antigen concentration [Ag]≈K_D_, which substituted into Equation (2) gives
[Ag] ≈ exp(∆G/k_B_T) (5)

Thus, the model suggests an exponential relationship between the immunologically adjusted free antigen concentration [Ag] and the binding free energy. Since the availability of antigen corresponds to the probability of an interaction, expansion of the antibody system in the proposed interaction space towards antigen shape space will also be exponentially distributed. Availability of antigen here means the concentration of unbound antigen (epitope, more strictly speaking) that is available for triggering B cells and that has been adjusted by B-cell differentiation, antibody production and antigen removal by FcR-bearing effector cells, defined as global antibody equilibrium [12].The combination of Gaussian-distributed interactions of random structures with exponentially distributed Ag frequencies yields a lognormal distribution of interaction free energy levels (Figure 5).

Consequently, a normal distribution with mean interaction energy μ = 0 and variance σ^2^ = RT^2^ is transformed into a lognormal distribution with mean = exp(μ + σ^2^/2). This approach gives a mean energy of 27.67 kJ/mol, which is equivalent to a K_D_ of 2 × 10^−5^. This is in a good agreement with the observed lower limits of antibody-binding energy and is also the lower limit of BCR sensitivity [59].

In order to define the behavior of a system of molecules, we can introduce a system equilibrium constant
K_sys_ = K_A_/<K_A_> = exp(^−^(∆G^−^<∆G>)/k_B_T)(6)
where < > enclose median equilibrium constant and mean free energy of binding in the system of antibodies, respectively. K_sys_ represents the binding propensity of a given molecule in the system. The reason for introducing this parameter is that although in a bimolecular interaction, K_A_ is a sufficient measure for determining the ratio of bound and free molecules in equilibrium, in a system, a given molecule can interact with any other component of the system and its relative free energy will determine its behavior in the system. Instead of looking at particular bimolecular interactions, we are interested in the flow of antigen arising from the sequential interactions with antibodies with increasing affinity.

To further explore the properties of the system, we shall consider the combination of a lognormal distribution and an exponential distribution by exponential sampling, as has been described by Reed and Mitzenmacher [60,61]. For the whole system, we need to consider the exponentially distributed number of normal distributions of interaction energy, as explained above, corresponding to exponentially distributed number of lognormal distributions of equilibrium constants. The probability density function of K_sys_ with such distribution is given by Equation (7).
(7)$$f\left(K_{sys}\right)=\int_{r=0}^{\infty }-\lambda e^{\lambda r}\frac{\sqrt{\lambda }}{\sqrt{\pi r}2\Delta EK_{sys}}e^{-\frac{\lambda {\left(\Delta G_{sys}\right)}^{2}}{4r\Delta E^{2}}}\;dr$$
using a λ rate of exponential decrease of antigen concentrations and a variance of 2/λ for the normal distribution. We can regard the energy fluctuation associated with ligand-independent tonic signaling [62] as the unit variance ∆E^2^.

This combination of exponential and lognormal distributions generates a double-Pareto distribution [61] of K_sys_, which for K_sys_>1 is given by
f(K_sys_) = λ/2 *K_sys_^−1− λ^(8)
which is a power law distribution of system affinity constants with degree exponent λ+1 (Figure 5 and Figure 6). From this distribution, we can now obtain the network description of a system of antibody interactions. A more detailed description of the deduction of power law distribution and the interpretation of interaction space is given in Appendix A.

### 4.2. Hierarchy of Binding Events and Geometry of the System

Following an active expansion phase of the immune response, when a number of short-lived plasma cells is generated and antigen is cleared by means of the antibodies produced, the immune response settles and the system retracts. MBC and LLPC are formed, which provide walls of protection against future intrusion by that particular antigen [63]. While MBC can enter GC reaction upon antigenic rechallenge, LLPC are terminally differentiated. This also means that while MBC are dependent on tonic BCR signaling, LLPC are independent of such signals.

In the RADARS model, these walls of protection can be thought of as an outermost surface of high-affinity antibodies produced by LLPC and an inner multiple layer of MBC. The affinity of the antibody towards cognate antigen determines the distance of the surface of protection in that particular direction in interaction space. This translates to a maximal concentration of that antigen, since high-affinity antibodies can reduce target concentration with an efficiency determined by affinity. Should antigen concentration rise further, MBC of lower energy layers are triggered to enter germinal centers again, expand and differentiate, producing new antibody-secreting cells and a stronger wall of protection [55].

The important question here is the organization of this hierarchy. Both MBC and LLPC may persist for a lifetime [58,64], so the optimal coverage of shape space in balance with affinity requires organization of interaction energy and space. In the case of LLPC, we know of nonantigen-specific deletion mechanism, so competition for niche, which is not antigen specific, might be the only attrition factor [65,66]. MBC may undergo further rounds of affinity maturation in GC and can therefore be reorganized [67]. The affinity of antibodies produced by LLPC is generally expected to be high [68], though initial low affinity of BCR favors differentiation into LLPC following GC reaction, while initial high affinity has opposite effects [69]. This observation underlines that hierarchy needs to be established rather than single high-affinity cells selected during an immune response.

### 4.3. A Scale-Free Network of Interactions

The systems organization of binding events, as outlined above, can be interpreted as a network of interactions. While antibodies seek to minimize their free energy by finding their best fitting target, antigens are passed on from antibody to antibody. Such shared consecutive binding represents links in the network between nodes of antibodies. Antibodies with the highest K_sys_ value are the most avid binders and as such will take over and handle most antigens, which are channeled to these molecules by antibodies underneath in the binding hierarchy. These antibodies will have the highest number of links and therefore, the highest network degree k. The RADARS model in combination with the GQM [29] suggests that long-lived plasma cells act as network hubs. By constantly secreting antibodies, these terminally differentiated cells provide for the binding and removal of antigen of all network nodes, represented by memory B cells, below in the hierarchy. This prevents activation of memory B cells and maintains their resting state. At the bottom of the hierarchy, B1 cells, producing natural antibodies, serve as a first line of defense and relay agents for antigen (Figure 6). The higher the network degree and corresponding K_sys_, the more cells and structure space is covered by a plasma cell.

Assuming that k = K_sys_, we obtain the probability density function of the antibody network
p(k) ~ k^−γ^(9)
where γ is the degree exponent and is equal to λ+1 (Figure 5).

Assuming that each subnetwork of LLPC has a similar organization determined by thermodynamic laws, the whole system is expected to have a self-similar fractal topology [70]. Fractality in such networks is characterized by the box dimension dB, which is related to the degree exponent [71] by
d_B_ = (γ^−^1)/(γ^−^2)(10)

The network degree distribution exponent in the RADARS model describes the connectivity between shells of free energy levels; on the other hand, fractal box dimension describes the rate of appearance of clusters in the network projected onto the surface of the system. Therefore, a further level of similarity can be introduced if these two parameters are equal. An ideal network with equal network degree distribution exponent and fractal dimension is satisfied by one condition only, if d_B_ = golden ratio + 1 since φ/(φ^−^1) = φ^2 =^ φ + 1 (golden ratio:φ = (1 + √5)/2). It is proposed that an optimal resting immune network in thermodynamic equilibrium could be characterized by this ideal network with degree distribution exponent γ = φ + 1 = φ^2^. Then the probability that an antibody can cross-react with k other antibodies will decay as a power law p(k) ~ k^−γ^ and the number of boxes N_B_ needed to tile a renormalized network, i.e., the probability of finding antibody interaction clusters characterized by the shortest path l_B_, will decay as a power law N_B_(l_B_)/N ~ l_B_^−γ^. In our model, l_B_ is in fact a distance along the projected surface of the system and as such, it represents level of structural similarity.

The power law relationship of antibody interactions is a hallmark of scale-free networks [72]. This scale-free network is an energy transfer system physically and an antigen transfer system immunologically. This is an optimization of antibody differentiation in the sense that the minimal number of high free energy antibodies (network hubs) are used for the removal of the maximal diversity of antigens, covering the maximum of immunologically relevant structure space. The generation of an antibody network, with network hubs represented by plasma cells secreting antibodies, reveals the physical aspect of the system: all interactions of such an antibody contribute to the clearance of many target antigens sharing structural homology. A new node in the network, a new B cell in the structure space, will preferentially attach to an existing subnetwork as a low-affinity clone, in agreement with the preferential attachment model of growth in scale-free networks [73]. Such preferential attachment may explain immunodominance and antigenic sin, phenomena arising from the preference of the immune system for known epitopes, which correspond to directions of hubs in the network.

## 5. Steps towards Validation and Application of the Model

Absolute numbers of B cells and proportions of B-cell subpopulations are in agreement with the proposed model. Figure 5 and Figure 6 suggest that most of the antibody conformational diversity is present in the transitional and naïve B-cell compartment, followed by MBC and LLPC. With the development of the immune system, and with the aging of the host, bone marrow output of antigen inexperienced B cells decreases and the proportion of MBC increases [74]. Meanwhile, the number of LLPC in the body is several orders of magnitude lower [75].

The network architecture of antibody repertoires was recently computed based on high-throughput sequencing data from more than 100,000 unique antibody sequences [76]. This study revealed that pre-B cell and naïve B-cell clones form homogenously interconnected assortative networks, in contrast to the disassortative networks of plasma cell clones, which covered smaller but more focused regions of sequence space. This contrasting behavior of antigen-naïve and antigen-experienced, post-GC B cells corresponds to the antibody-centric network view in our model. The low-affinity region with developing B cells is homogenously interconnected by clonal relationships and shared usage of gene segments (Figure 2 and Figure 3). The high affinity side of the distribution is the narrowing, focusing interaction space of plasma cells, where growth of subnetworks with different specificity into different directions of antigen shape space appears as hub repulsion and fractal topology.

Considering that our technological capability is ripe for the high-resolution determination and comprehensive analysis of antibody sequence space, current efforts focus on the conversion of sequence space data into datasets in interaction space. By providing a physical and mathematical description of relationship between antibody clones, the RADARS model may help in the final integration of sequence data. The model also suggests that sequence-based network properties of early B-cell developmental stages also need to be determined, in addition to the mature and antigen-experienced repertoire, and comprehensive and selective analysis of the B1 repertoire is very important for capturing network properties of the system. Furthermore, predictions of antibody affinity [77,78,79] and of binding energy changes upon mutation [80] should be included in the algorithms to convert repertoire sequence data to interaction networks.

## 6. Conclusions

Network theory has always been considered as a key to understand and define immunity on a systems level. The network hypothesis of Niels Jerne [25], its modified version leading to the concept of clonal selection [81], mathematical and computational simulations [82,83], various reinterpretations [26], and experimental approaches using NGS or antigen microarrays [24,27,33] all strive to describe this highly complex system as connected elements of a network. There are two new aspects of the RADARS model that may improve our view of this network. First, it introduces physical units, binding energy, as the measure of interactions and as a measure of system architecture. Natural networks are formed as a result of energy dispersal [84,85], therefore network theories should consider energy transduction in the system. Second, it proposes an architecture for the whole network, characterized by the scale-free distribution, and an optimal value for the degree exponent of power law relationship.

The model presented here with a network degree exponent φ^2^ depicts an ideal state of the system of antibody interactions. It is the fluctuations and disturbances in the system that we observe as immune response during infections, and distortions are autoimmunity and allergy. Besides suggesting how antibody sequence space fits into structural space and into an interaction network, the model may potentially lead to the ability to model whole immune system and simulate its functioning. The RADARS model proposes that a universal organization of an immense number of structures in a huge but finite system is possible by adjusting the resolution of structural recognition, which is the adjustment of interaction energy. Radial adjustment of system resolution generates a network of interactions, which is scale-free and is characterized by a power law distribution of free energy of interactions. Overall, this organization allows the energy-optimized controlled removal of antigens from the host system.

## Figures and Tables

**Figure 1 antibodies-09-00013-f001:**
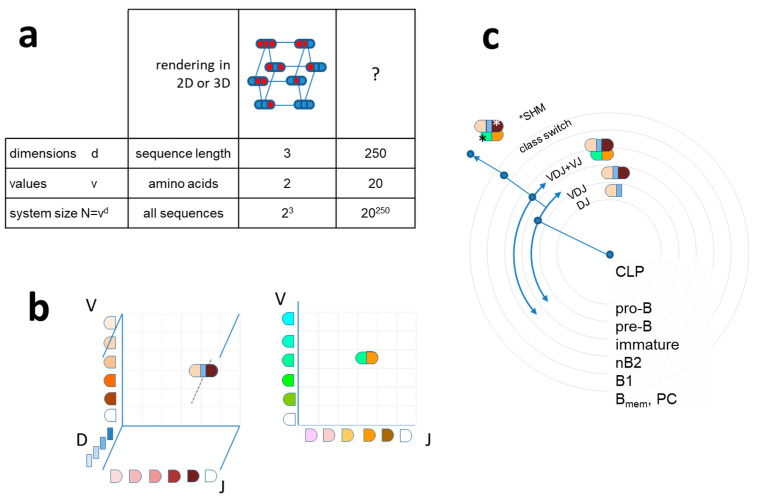
Sequence space and visualization of antibody sequence relationships (**a**) Theoretical diversity (system size N) of a sequence is determined by its length (dimension) and the number of values a particular position in the sequence can take. An antibody Fv region of 250 amino acids has an astronomical sequence diversity if full randomization is allowed. (**b**) Antibody sequences are frequently interpreted as recombined germline sequences. Such display of combinatorial diversity may allow the tracking of specific clonal expansions and further diversification by somatic hypermutation (SHM). (**c**) The potential development scheme of a given antibody clone is shown with antibody sequence development along with B-cell differentiation steps. Arching arrows represent combinatorial diversification by V–D–J rearrangement and light chain paring. CLP, common lymphoid progenitor; nB2, naïve B2 cell; Bmem, memory B cell; PC, plasma cell.

**Figure 2 antibodies-09-00013-f002:**
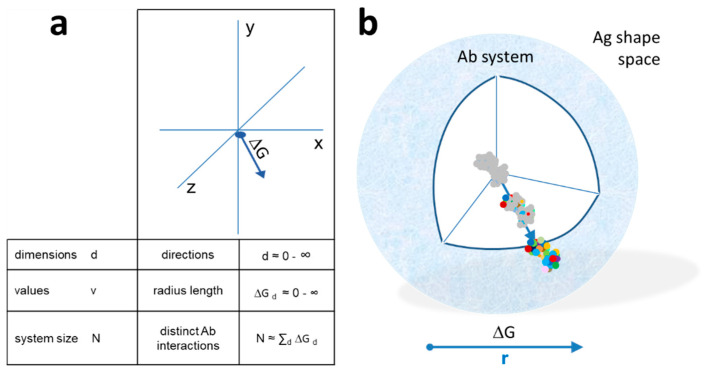
Quantitative interaction space and B-cell differentiation. (**a**) The system of antibodies has a center as a reference point in a conceptual three-dimensional metric space of interactions. Structural diversity appears as directions (exemplary arrow) in Ag shape space. Distance is measured as the free energy change of antibody–antigen interaction. The total size of the system N is the sum of all interactions in all directions. *(***b**) Diversity appears as we leave the center of the system, spherical shells representing various levels of resolution of molecular recognition. Colors represent distinct amino acids of antibody-binding site engaging in noncovalent bonding with the target molecule. Higher energy interactions engage more residues.

**Figure 3 antibodies-09-00013-f003:**
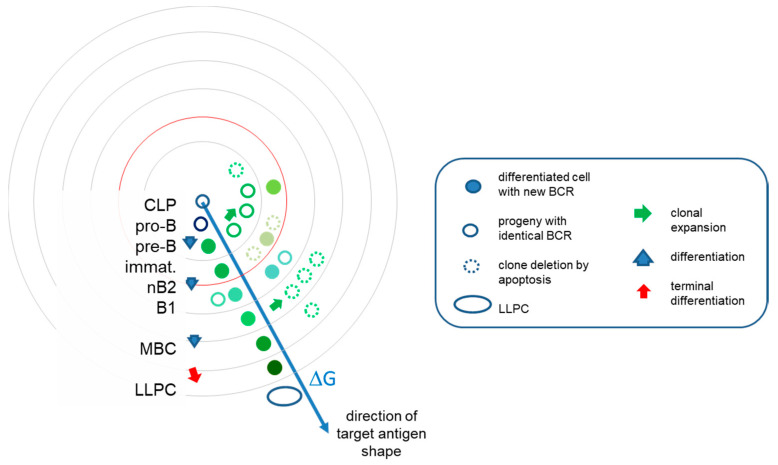
B-cell development in interaction space. The evolution of the system of antibodies can be interpreted as clones filling the interaction space at various levels of resolution. Along this pathway, cells continually increase their BCR affinity towards the target. Beyond the mean energy of the system, represented by red circle, TD responses allow further directed differentiation via somatic hypermutations in germinal centers. Clonal expansions are however followed by retraction, effector cells eliminated by apoptosis and only memory B cells and plasma cells surviving. Different colors stand for structural differences and relationships. LLPC, long-lived plasma cell; MBC, memory B cell; immat, immature B cell.

**Figure 4 antibodies-09-00013-f004:**
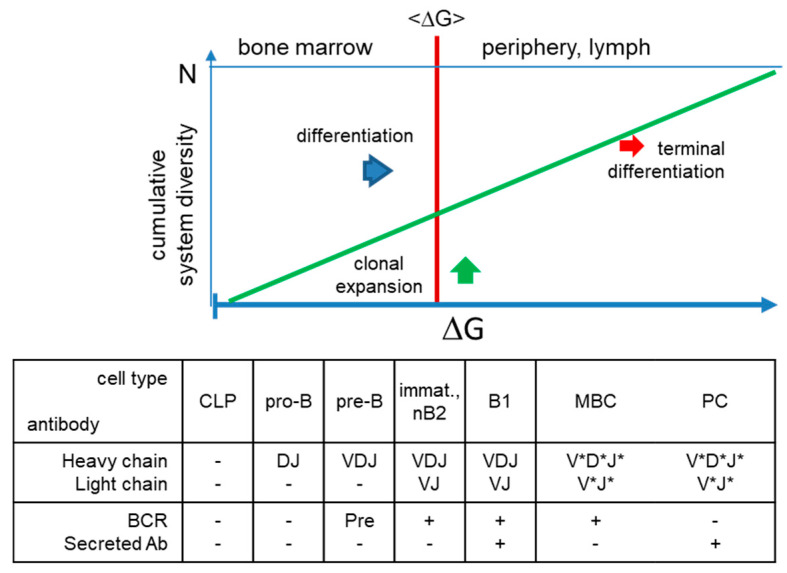
Generation of structural diversity in the humoral immune system. Beyond the bone marrow, multidirectional growth towards antigenic shapes generates diversity in the system, further enlarging that of the naïve repertoire. The energy of these interactions is adjusted for each antigen based on its immunological properties, thereby secreted antibodies can adjust the concentration of the target antigen in the host. *somatic hypermutations; CLP, common lymphoid progenitor; immat., immature; nB2, naive B2 cell; MBC, memory B cell; PC, plasma cell; BCR, B-cell receptor.

**Figure 5 antibodies-09-00013-f005:**
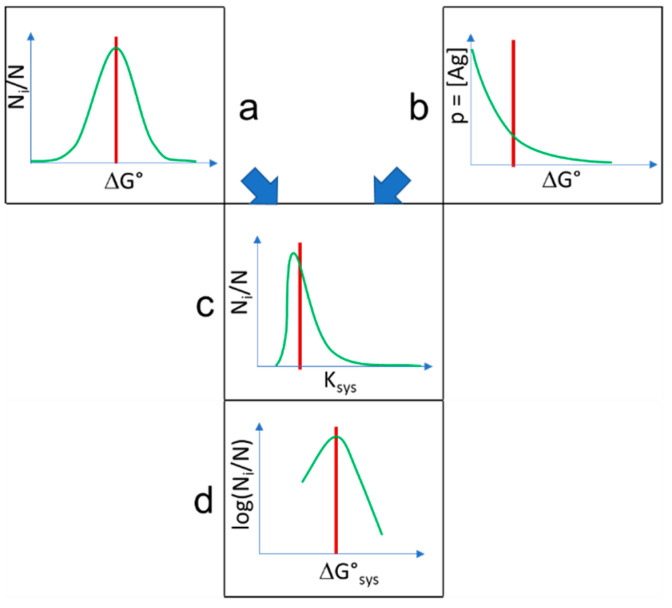
Combination of distributions. Normally distributed interaction energies (**a**) are selected by exponentially distributed probability of interactions with antigen (**b**), resulting in double-Pareto distribution of system equilibrium interaction constants (**c**), which corresponds to the distribution of conformational entropy in interaction energy levels (**d**) when logarithm is taken. Red lines stand for the expected values of the distributions. N_i_/N is the relative conformational diversity normalized to system size N.

**Figure 6 antibodies-09-00013-f006:**
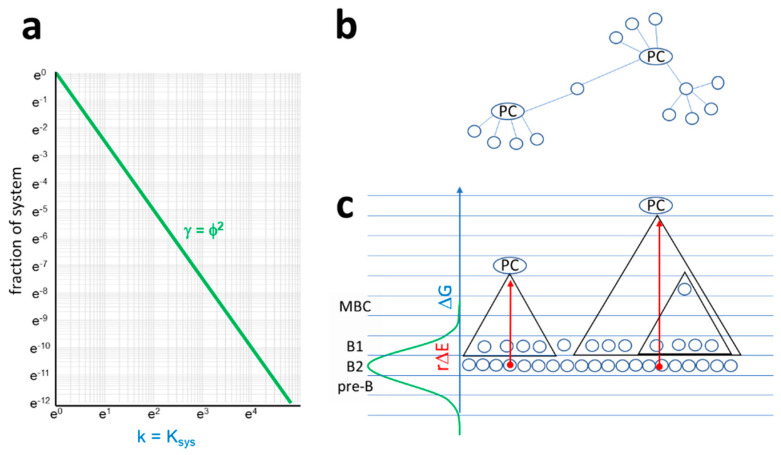
Properties of the antibody interaction network. (**a**) In the antibody network, a node is a B cell at a particular differentiation stage and a given BCR sequence. Links are pathways of antibody–antigen interactions, along which antigen is transferred from a lower energy node to a higher energy node. Degree distribution of the antibody network corresponds to the probability distribution of relative free energy of antibody interactions, where the system equilibrium constant is the degree of node and Ni/N is relative frequency of nodes with that degree in the interaction network. (**b**,**c**). A schematic network organization of antibody interactions is proposed. In subnetworks, plasma cells act as hubs, being connected to all memory B cells and B1 cells with shared structures at lower free energy levels. By secreting antibodies, plasma cells control activation of all connected cells. Triangles represent subnetworks, blue lines indicate energy levels, and red arrows stand for energy deviation.

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
