# Peer review of "Network Organization of Antibody Interactions in Sequence and Structure Space: the RADARS Model"

_2073-4468, 2020, doi:10.3390/antib9020013_

Round 1

Reviewer 1 Report

The author presented an interesting concept paper of RADARS model, which deals with network organization of antibody interactions in sequence and structure space. This is quite a novel study, and will attract wide attention from public. I suggest addressing following minor issues.

  1. Methods of related calculation and concept development should be described in details. Right now it's quite lacking.
  2. It's suggested to include a session to verify the RADARS model. For example, please cite some related references and validate your model.

Author Response

I wish to thank the reviewer for undertaking this assignment and for the critical and helpful suggestions. My detailed answers are below. Because of the corrections the line numbers have shifted, so the new positions of the corrected lines I indicated in parenthesis.

  • "Methods of related calculation and concept development should be described in details. Right now it's quite lacking."

  • The model heavily builds on an interconnected set of assumptions, which are described in detail in previous publications. This is indicated in lines 89-90, where these publications are cited. Nevertheless, the best efforts were made to explain that approach to the readers of this paper, and hopefully the specific changes upon the reviewers’ suggestions also help in this aspect. All of the equations and functions shown in the paper are accompanied by citations of papers where these are described in detail. The most important equation of the paper, equation 7, has a supplementary online text to help the interested readers follow the details of deducing the power law function in equation 8 from the integral of equation 7.
  • "It's suggested to include a session to verify the RADARS model. For example, please cite some related references and validate your model."

  • Section 5 was renamed (lines 454-455) and reorganized, with new citations inserted, to highlight that the model is in agreement with numbers of B-cell subpopulations in general and also with current sequence-based findings of systems-scale antibody networks, but further work is warranted for validation and application of the model. A critical part in applying the RADARS model to repertoire sequence data is the prediction of antibody affinity, thereby placing a particular sequence into a particular level in the binding energy hierarchy. This step is in fact so important that its optimization, comparison of different methods, merits a scientific article in itself. I am actively looking for cooperations to realize these steps in the near future.

Extra changes

Line 341: missing parenthesis in equation 6 was added.

Line 310: abbreviation MBC was corrected

Reviewer 2 Report

Comments to article antibody-761254

The proposed RADARs mathematical model offers a new opportunity to describe antibody interaction space and network. Moving away from sequence analysis, the authors look at antibody-antigen interactions as the interaction between two structures with particular surface charge and hydrophobicity distribution, hence focusing on their interaction energy. Other key parameter is the probability for such interaction to occur. These physical properties are preferred to the standard sequence analysis as they allow to mathematically describe antibody interaction space and network. Also, they help against the scarcity of protein structures available compared to the multitude of antibody sequences.

Beside its intrinsic value, this quantitative description of the system could help processing the vast amount of Ab sequences available for structural data prediction.

General note:

The article has been reviewed with particular attention to the statements and the theoretical assumptions at the base of the proposed model, rather than verifying the exactness of the mathematical equations used to describe the specific immunological processes addressed by the model.

Major comments:

1) RADARs is presented as an alternative to sequence based analysis to describe antibody interactions. One of the main drivers behind RADARs is the low number of antibody structure available to be associated to a given sequence for the analysis. Yet, there are some excellent exceptions to it. The application of RADARs to a well described scenario (e.g. Burnett et al. 2018 10.1126/science.aao3859., or formation of autoimmune antibodies, vaccination, etc..) would be of great interest for the community and a great added value for this manuscript.

2) Despite a model by definition needs to include some simplifications to be able to embrace several scenarios, it would be appropriate to briefly mention which are the simplifications adopted. For example, the model does not take in account antigen immunogenicity. Only to cite one aspect of it, antigens that show high homology to self proteins have less immunogenic epitopes. Other example: the interaction between antibody and effector cells is also neglected.

Minor comments:

1) Check for small English grammar oversights

2) Lanes 29-30: I associate these words to the organization of cells into tissues and organs, rather than the main driving behind the genesis of the adaptive immune system. With “complex multicellular life” is it maybe meant vertebrates?

2) Lanes 33-34: does it refer to humans specifically?

3) Lanes 51-54: maybe in this perspective it is wroth to mention other technologies that well complement antigen array screenings, e.g. antigen phage/bacteria display. At least two approaches are used in this context: synthetic peptide libraries or whole genome/ORFeome libraries screening onto immobilized patient/donor serum (Wu et al. 2019 (doi: 10.1074/mcp.RA119.001582.); Zantow et al. 2018 (doi: 10.1007/978-1-4939-7447-4_27.)).

4) Lane 74: Ab-Ag interaction and Ab-effector cell interaction

5) Lanes 77-79: yes, but the structure is determined by the sequence. In the end, also the physical properties can be referred to the sequence. I do not think it can be said that one approach better defines the system than the other, as they can complement and integrate each other.

6) Lane 118: somatic hypermutation (SHM)

7) Lanes 123-124: the message delivered with this wording way may be misleading. The fact that “HCDR3 is necessary, albeit insufficient, for specific antibody binding” (ref. 37 abstract), because specific target binding depends also on VL pairing and specific VDJ rearrangement, does not mean that the same HCDR3 is often seen in different antibodies with different antigen specificity. When using only one sequence per Ab for classification, than the HCDR is still the most appropriate.

8) Lanes 306-312: it should be considered that the [available Ag to the Ag presenting cells] differs from the [Ag in the body]. Also immunogenicity varies from protein to protein. Protein size is a factor. Antigen accessibility. This implies that the sentence: “the availability of antigen corresponds to the probability of an interaction”, should be revised in this context.

Author Response

I wish to thank the reviewer for undertaking this assignment and for the critical and helpful suggestions. My detailed answers are below. Because of the corrections the line numbers have shifted, so the new positions of the corrected lines I indicated in parenthesis.

Major comments

  • "RADARs is presented as an alternative to sequence based analysis to describe antibody interactions. One of the main drivers behind RADARs is the low number of antibody structure available to be associated to a given sequence for the analysis. Yet, there are some excellent exceptions to it. The application of RADARs to a well described scenario (e.g. Burnett et al. 2018 10.1126/science.aao3859., or formation of autoimmune antibodies, vaccination, etc..) would be of great interest for the community and a great added value for this manuscript."

  • Section 5 was renamed (lines 454-455) and reorganized, with new citations inserted, to highlight that the model is in agreement with numbers of B-cell subpopulations in general and also with current sequence-based findings of systems-scale antibody networks, but further work is warranted for validation and application of the model. A critical part in applying the RADARS model to repertoire sequence data is the prediction of antibody affinity, thereby placing a particular sequence into a particular level in the binding energy hierarchy. This step is in fact so important that its optimization and comparison of different methods, merits a scientific article in itself. I am actively looking for cooperations to realize these steps in the near future.
  • "Despite a model by definition needs to include some simplifications to be able to embrace several scenarios, it would be appropriate to briefly mention which are the simplifications adopted. For example, the model does not take in account antigen immunogenicity. Only to cite one aspect of it, antigens that show high homology to self proteins have less immunogenic epitopes. Other example: the interaction between antibody and effector cells is also neglected."

  • As the reviewer remarks, models require simplifications, and this model makes several assumptions as well. In fact, the model heavily builds on an interconnected set of assumptions, which are described in detail in previous publications. This is indicated in lines 89-90, where these publications are cited. Nevertheless, the best efforts were made to explain that approach to the readers of this paper, and hopefully the specific changes upon the reviewers’ suggestions also help in this aspect.

Minor comments

1, Spelling and grammar was corrected by applying US English correction in Word.

2a, Lanes 29-34 were rephrased, now referring first to immunity in general, then to the adaptive immune system of vertebrates.

2b, Yes, the numbers are for humans, which is now mentioned explicitly. (line 37)

3, The text was complemented with the suggested technological approaches and references. (line 60-61)

4, I agree that understanding and incorporating interactions of antibodies with effector cells are indispensable for generating useful models, however I cannot yet see how that could be done in a quantitative manner for networks. Therefore, I am not referring to this aspect of the interactions in this particular sentence, this introductory part rather intends to draw attention to what the paper is going to discuss.

5, These sentences I think are in agreement with the reviewers opinion, in the sense that the proposed interaction network is supposed to “help better define the system” rather than replace sequence-based approaches. I absolutely agree that sequence determines structure and currently NGS technologies are capable of generating reliable immunoglobulin data for predictions – which in turn could be fine-grained using different models, perhaps including the proposed RADARS model. I modified this sentence accordingly (line 87).

6, Abbreviation of SHM was explained in the text as suggested. (line 126)

7, I softened my criticism on the use of H-CDR3 for functional classification, as suggested by the reviewer. The corrected sentence is now in lines 130-133. “This region is regarded as being most important for determining binding specificity, but some precaution is necessary when only H-CDR3 is used for functional classification.”

8, Revision of the sentence “the availability of antigen corresponds to the probability of an interaction”.

Expressing the availability of antigen by numbers is indeed a difficult task and requires thorough description of what is exactly meant here. The model referenced here (GQM) assumes that free antigen concentration [Ag] is that available for triggering specific B cells. Immunogenicity in the model is composed of two factors: 1, concentration of antigen, which is activity in the biochemical sense, and availability to B cells in lymphoid organs in the immunological sense; 2, danger posed to the host, triggering all kinds of second signals, which keep affected B cells differentiating. [Ag in the body], as suggested by the reviewer, and interpreted as concentration of accessible antigen in the body, should be represented in the lymphoid organs if immunological surveillance is working (perhaps except for special, immunologically isolated organs, beyond barriers). [available Ag to the Ag presenting cells], interpreted as what is actually present in the lymphoid organs, could therefore be quite similar. It is important to note that the cited model assumes a global equilibrium of antibody homeostasis, meaning that the concentration of each antigen, meaning the antigen available for B cells, more specifically the concentration of each unbound epitope, has been adjusted both by the differentiation of relevant B-cell clones (adjusting KD and [Ab]) and by the effector cells bearing FcR and capable of removing bound Ag. A new sentence was added to explain this interpretation, lines 322-325.

Extra changes

Line 341: missing parenthesis in equation 6 was added.

Line 310: abbreviation MBC was corrected